# PASCAL: Precise and Efficient ANN-SNN Conversion using Spike Accumulation and Adaptive Layerwise Activation

**Pranav Ramesh**                                                     *cs22b015@smail.iitm.ac.in*
*Department of Computer Science and Engineering*
*Indian Institute of Technology Madras*

**Gopalakrishnan Srinivasan**                                        *sgopal@cse.iitm.ac.in*
*Department of Computer Science and Engineering*
*Indian Institute of Technology Madras*

**Reviewed on OpenReview:** *https://openreview.net/forum?id=kIdB7Xp1Iv*

## Abstract

Spiking Neural Networks (SNNs) have been put forward as an energy-efficient alternative to Artificial Neural Networks (ANNs) since they perform sparse Accumulate operations instead of the power-hungry Multiply-and-Accumulate operations. ANN-SNN conversion is a widely used method to realize deep SNNs with accuracy comparable to that of ANNs. Bu et al. recently proposed the Quantization-Clip-Floor-Shift (QCFS) activation as an alternative to ReLU to minimize the accuracy loss during ANN-SNN conversion. Nevertheless, SNN inferencing requires a large number of timesteps to match the accuracy of the source ANN for real-world datasets. In this work, we propose PASCAL, which performs ANN-SNN conversion in such a way that the resulting SNN is mathematically equivalent to an ANN with QCFS-activation, thereby yielding similar accuracy as the source ANN with minimal inference timesteps. In addition, we propose a systematic method to configure the quantization step of QCFS activation in a layerwise manner, which effectively determines the optimal number of timesteps per layer for the converted SNN. Our results show that the ResNet-34 SNN obtained using PASCAL achieves an accuracy of $\approx 74\%$ on ImageNet with a $56\times$ reduction in the number of inference timesteps compared to existing approaches. The code is available at https://github.com/BrainSeek-Lab/PASCAL.

## 1 Introduction

Deep Artificial Neural Networks (ANNs) have revolutionized the field of machine learning by achieving unprecedented success for various computer vision and natural language processing tasks (LeCun et al., 2015). Despite their superhuman capabilities, there exists a significant computational efficiency divide between ANNs and the human brain. In recent years, Spiking Neural Networks (SNNs) have emerged as a plausible energy-efficient alternative to ANNs (Roy et al., 2019; Tavanaei et al., 2019; Guo et al., 2023). The fundamental building block of an SNN, namely, the spiking neuron, encodes input information using binary spiking events over time. The sparse spike-based computation and communication capability of SNNs have been exploited to obtain improved energy efficiency in neuromorphic hardware implementations (Davies et al., 2021).

Training methodologies for SNNs can be classified primarily into two approaches, namely, direct training using spike-based algorithms and ANN-SNN conversion. The efficiency of these methods is compared in A.11. Direct SNN training using spike-based error backpropagation (Lee et al., 2016; Wu et al., 2018; Shrestha & Orchard, 2018; Neftci et al., 2019; Lee et al., 2020) has been shown to yield competitive accuracy on complex vision datasets such as ImageNet (Rathi et al., 2020; Deng et al., 2022). It is important to note that SNNs require multiple forward passes, as inputs are fed sequentially over a certain number of timesteps.

As a result, performing backpropagation on SNNs is both compute- and memory-intensive because of the need to accumulate error gradients over time. Despite being computationally prohibitive, backpropagation methods have been shown to yield competitive accuracy, with inference latencies typically less than a few tens of timesteps (Deng et al., 2022).

ANN-SNN conversion, on the other hand, eliminates the need for direct SNN training by using the pre-trained ANN weights and replacing the nonlinear activation (for instance, ReLU) with the equivalent spiking non-linearity (integrate-and-fire). The ANN activation is effectively mapped to the average firing rate of the corresponding spiking neuron. The conversion methods have been shown to provide accuracy comparable to that of the original ANN, albeit with longer inference latencies typically spanning hundreds to thousands of timesteps (Sengupta et al., 2019). This is because the correlation between the ANN activation and spiking neuronal firing rate improves with increasing number of timesteps during SNN inference (Cao et al., 2015). Several prior works have proposed techniques to optimize the number of inference timesteps (Diehl et al., 2015; Rueckauer et al., 2017; Sengupta et al., 2018; 2019; Han & Roy, 2020; Deng & Gu, 2021). Most notably, Bu et al. proposed a novel activation function, namely, the Quantization-Clip-Floor-Shift (QCFS) activation, as a replacement for the ReLU activation during ANN training. The QCFS function, with pre-determined quantization step $L$, was formally shown to minimize the expected ANN-SNN conversion error. We note that the QCFS activation works well in practice for CIFAR-10, but for CIFAR-100 and ImageNet, the ANN accuracy can only be matched when the corresponding SNN is inferred for $\sim$1024 timesteps despite being trained using only 4 or 8 quantization steps (Bu et al., 2023). In effect, the number of SNN inference timesteps needed to match the ANN accuracy depends on the complexity of the dataset.

Our work addresses this issue using a precise and mathematically grounded ANN-SNN conversion method, in which the number of inference timesteps depends only on the structure of the source ANN, and not on the dataset. We utilize the integrate-and-fire spiking neuron with soft reset (Rueckauer et al., 2017; Han et al., 2020), a technique that is critical for achieving near-zero conversion loss. Next, we address a key limitation of SNNs, which is their inability to handle negative inputs effectively, since spikes are commonly treated as binary values representing either 0 or 1. We demonstrate that **inhibitory (or negative) spikes** are essential for ensuring the mathematical correctness of our method. Finally, we propose an algorithm to determine the optimal quantization step for QCFS activation per layer by computing a statistical layer-sensitivity metric. This enables an optimal trade-off between computational efficiency and accuracy. In summary, we propose PASCAL, an accurate and efficient ANN-SNN conversion methodology, which has the following twofold contributions.

- **Spike-count and Spike-inhibition based SNN.** We develop a novel spike-count based approach, which is mathematically grounded and incorporates both excitatory as well as inhibitory spikes, to transform an ANN with QCFS activation into a sparse SNN. We realize ANN accuracy using fewer timesteps than state-of-the-art conversion approaches (refer to Section 4), thus lowering both inference time and energy consumption. We achieve near-lossless ANN-SNN conversion with a small additional overhead caused by introducing a spike counter per neuron.

- **Adaptive Layerwise QCFS Activation**. We propose a statistical metric to systematically analyze and categorize layers of a DNN to selectively use higher precision (or QCFS steps) for certain layers, and lower precision for others. The DNN layers with lower precision need to be unrolled for fewer timesteps, thereby, improving the computational efficiency during SNN inference. This is in contrast to early-exit SNNs, in which all the layers are executed for the same number of timesteps, which in turn depends on the input complexity (Srinivasan & Roy, 2021; Li et al., 2023b).

We validate the efficacy of PASCAL on CIFAR-10, CIFAR-100 and ImageNet datasets (described further in Appendix A.3), with both VGG-16 and ResNet architectures. In addition, we demonstrate the hardware efficiency of SNN, obtained with both uniform and Adaptive Layerwise (AL) activation.

Table 1: Summary of notations used in this paper.

| Symbol | Definition | Symbol | Definition |
|---|---|---|---|
| $l$ | Layer index | $X$ | Generic input matrix |
| $\widehat{h}$ | QCFS function used in ANN training | $s^l$ | Output spike train of layer $l$ |
| $\widetilde{h}$ | Integrate-and-Fire (IF) layer used in PASC formulation | $W$ | Weight matrix |
| $x$ | A single element in $X$ | $H(\cdot)$ | Heaviside step function |
| $z^l$ | Weighted output after layer $l$ | $z_i^l$ | $i^{th}$ unrolled timestep of the weighted output after layer $l$ |
| $\theta_n$ | Threshold of current IF layer | $L_n$ | Quantization Step for current IF layer |
| $s$ | Spike count | $L_{n-1}$ | Quantization Step for previous IF layer |
| $\lambda^l$ | Trainable threshold in ANN for layer $l$ | $\varphi$ | Shift term of QCFS activation |
| $L$ | Generic quantization step | $mem(t)$ | Membrane Potential after $t$ timesteps |
| $b$ | Bias term | $\mu$ | Batch mean |
| $\sigma^2$ | Batch variance | $\epsilon$ | Constant for numerical stability of batch normalization operation |
| $\gamma$ | Scaling parameter for batch normalization | $\beta$ | Shifting parameter for batch normalization |
| $z$ | Input to a classification model | $\mathbf{C}$ | Set of output classes |
| $*$ | MatMul operation for a convolution or fully-connected layer | $f_{\mathcal{M}}^z$ | Classification map for a classification model $\mathcal{M}$ with input $z$ |
| $T$ | Inference timesteps | $\mathbf{L_{tot}}$ | Total number of layers |

## 2 Preliminaries and Related Work

### 2.1 ANN-SNN Conversion

ANN-SNN conversion has been shown to be a promising approach for building deep SNNs yielding high enough accuracy for complex visual image recognition (Cao et al., 2015; Diehl et al., 2015; Pérez-Carrasco et al., 2013; Rueckauer et al., 2016; 2017; Sengupta et al., 2019) and natural language processing tasks (Diehl et al., 2016). The conversion algorithms incur the following steps.

- The trained weights of the ANN are transferred to an architecturally equivalent SNN.

- The ANN activation is replaced with an appropriate spiking activation. In discriminative models, ReLU is replaced with Integrate-and-Fire (IF) spiking activation. An IF neuron integrates the weighted sum of inputs into an internal state known as the membrane potential, as described in Section 3. It emits an excitatory (or positive) spike if the membrane potential exceeds a predetermined firing threshold. The membrane potential is subsequently reset to a suitable value.

Efficient ANN-SNN conversion requires careful threshold initialization at every layer of the network so that the spiking rates are proportional to the ReLU activations. Diehl et al. proposed a model-based threshold balancing scheme, wherein the firing thresholds are determined solely based on ANN activations. Sengupta et al. proposed an improved data-based conversion scheme that additionally uses SNN spiking statistics for threshold initialization. However, the aforementioned methods were susceptible to information loss during SNN inference due to the use of *hard reset* spiking neurons, wherein the membrane potential is reset to zero in the event of a spike. Han et al. demonstrated near lossless low-latency ANN-SNN conversion with *soft reset* neurons, wherein the potential is decreased by an amount equal to the neuronal spiking threshold. Subsequently, Bu et al. showed that the threshold voltage could be learned, instead of configuring it heuristically post training, by replacing ReLU with the Quantization-Clip-Floor-Shift (QCFS) activation, which further minimizes the conversion error.

### 2.2 Temporally Efficient SNN Inference

In parallel with improving the ANN-SNN conversion process, there have been efforts to speed up SNN inference by optimizing the number of timesteps needed to achieve the best accuracy. Early-exit inference

methods, which incorporate auxiliary classifiers at various stages in the model, have been shown to improve the inference efficiency of both ANNs and SNNs (Panda et al., 2016; Huang et al., 2018; Srinivasan & Roy, 2021). Li et al. proposed a unique early-exit approach for SNNs, wherein the number of inference timesteps is modulated based on the input complexity. In this work, we propose a novel methodology to achieve both precise and temporally efficient ANN-SNN conversion.

## 3 Methodology

### 3.1 Quantization-Clip-Floor-Shift (QCFS) Activation

The QCFS function, $\widehat{h}$, as proposed by Bu et al., can be formulated as

$$\widehat{h}(\boldsymbol{z}^l) = \lambda^l \ \text{clip}\left(\frac{1}{L}\left\lfloor \frac{\boldsymbol{z}^l L}{\lambda^l} + \boldsymbol{\varphi} \right\rfloor, 0, 1\right), \tag{1}$$

where $z^l$ is the input to the activation layer, $L$ is the quantization step, and $\lambda^l$ is the trainable parameter for the activation threshold. At the end of training, $\lambda^l = \theta_n$, where $\theta_n$ is the final threshold that is subsequently used for SNN inference. Bu et al. showed that the expectation of conversion error reaches **0** when the shift amount, $\boldsymbol{\varphi}$, is $\frac{1}{2}$.

### 3.2 Precise ANN-SNN Conversion (PASC)

After obtaining a trained ANN, we replace the QCFS function with Integrate-and-Fire (IF) activation and transform the Matrix Multiplication (MatMul) layers for SNN inference so that model computation is unrolled over timesteps. The operations of the transformed IF and MatMul layers are described below.

**Integrate-and-Fire (IF) layer.** We consider the general case where the quantization step $L_{n-1}$ of the previous layer is different from that of the current layer ($L_n$). We initialize the layerwise threshold $\theta^* = \frac{\theta_n}{L_n}$, where $\theta_n$ is obtained from ANN training (refer to Section 3.1). The initial membrane potential is set to $\frac{\theta^*}{2}$, similar to prior works (Bu et al., 2023). We outline the procedure in Algorithm 1 for a generic layer, and Algorithm 2 for the input layer. We formally prove in Section 3.3 that the proposed algorithm enables precise ANN-SNN conversion. The algorithm for a generic layer is described below.

1. For the first $L_{n-1}$ timesteps, we perform the IF operation (detailed in Section 2.1) using the spike train obtained from the preceding layer. We use a spike counter to record the number of spikes produced at this stage.

2. In the next stage, we re-initialize the membrane potential to the value obtained at the end of the previous stage. Similarly, we set the initial spike count for this stage to that obtained from stage 1. We then perform the IF operation for the next $max(L_{n-1}, L_n) - 1$ timesteps. It is only at this stage that we introduce **inhibitory spikes**. If the membrane potential is less than zero, we emit an inhibitory spike and update both the membrane potential and the spike count accordingly.

3. For the final stage, we reset the membrane potential to the spike count obtained after stages 1 and 2. We then perform the IF operation for $L_n$ timesteps to generate the final spike train. The spike output of each timestep is stacked to create the final output tensor.

The output spike train generated by the PASC algorithm is effectively unrolled across $L_n$ timesteps by the subsequent layer. Therefore, we preserve the fact that after each layer, the number of inference timesteps is equal to the value of the quantization step, i.e. $T = L_n$. We note that Algorithm 2 is similar to, albeit simpler than, Algorithm 1, except that it receives the input image containing real-valued pixels (rather than binary spikes) over time. Feeding the input image to the SNN, without encoding the pixel intensities as spike trains, has been adopted in prior works to minimize accuracy loss during inference (Li et al., 2023b; Bu et al., 2023).

**MatMul layers.** The MatMul layer performs either a convolution or a fully-connected matrix multiplication, followed by batch normalization. These operations are generally performed using Multiply-and-Accumulate

---

**Algorithm 1** PASC Algorithm for Generic IF Layer

---

1: **Input:** An $L_{n-1} \times N \times H \times W$ matrix $\mathbf{X}$, where $L_{n-1}$ and $\mathbf{X}$ are the quantization step and output of the preceding MatMul layer, respectively.
2: **Output:** An $L_n \times N \times H \times W$ matrix representing the spike output (scaled by threshold) of the current layer.
3: **Initialization:** Threshold, $\theta^* = \frac{\theta_n}{L_n}$. Initial membrane potential, $mem(0) = \frac{\theta^*}{2}$. Initial spike count, $s = 0$.
4: **for** $t \in \{1, 2, \ldots L_{n-1}\}$ **do**
5:    $mem(t) = mem(t-1) + X[t]$
6:    **if** $mem(t) \geq \theta^*$ **then**
7:       $s = s + \frac{\theta_n}{L_n}$ // Increment on Excitatory spike
8:       $mem(t) = mem(t) - \theta^*$ // Soft reset upon spike
9: Set $mem(0)$ as the final membrane potential after the previous loop.
10: **for** $t \in \{1, 2, \ldots max(L_{n-1}, L_n) - 1\}$ **do**
11:    **if** $mem(t) \geq \theta^*$ **then**
12:       $s = s + \frac{\theta_n}{L_n}$ // Increment on Excitatory spike
13:       $mem(t) = mem(t) - \theta^*$
14:    **else if** $mem(t) < 0$ **then**
15:       $s = s - \frac{\theta_n}{L_n}$ // Decrement on Inhibitory spike
16:       $mem(t) = mem(t) + \theta^*$
17: Initialize a new SNN with threshold $\theta^* = \frac{\theta_n}{L_n}$, and initial membrane potential $mem(0) = s$, where $s$ is the spike count (scaled by $\theta^*$) computed previously.
18: Initialize an empty tensor $s^l$, with dimension $L_n \times N \times H \times W$, to generate the final spike train.
19: **for** $t \in \{1, 2, \ldots L_n\}$ **do**
20:    $s^l(t) = H(mem(t-1) - \theta^*) \cdot \theta^*$, where $s^l(t)$ is the spike output scaled by $\theta^*$, and $H$ is the Heaviside step function.
21:    $mem(t) = mem(t-1) - s^l(t)$
22: **return** $s^l$, which will be the input for the subsequent MatMul layer.

---

**Algorithm 2** PASC Algorithm for the Input Layer

---

1: **Input:** An $N \times H \times W$ input matrix $\mathbf{X}$.
2: **Output:** An $L_n \times N \times H \times W$ matrix representing the spike output (scaled by threshold) of the input layer.
3: Perform the computation as in an ANN, yielding the result $\widehat{h}(X) = \theta_n \, \text{clip}\left(\frac{1}{L_n} \left\lfloor \frac{X}{\theta_n} + \boldsymbol{\varphi} \right\rfloor, 0, 1\right)$.
4: Initialize an SNN with threshold, $\theta^* = \frac{\theta_n}{L_n}$. Initial membrane potential, $mem(0) = \widehat{h}(X)$. Initialize an empty tensor $s^l$ with dimension $L_n \times N \times H \times W$ to generate the final spike train.
5: **for** $t \in \{1, 2, \ldots L_n\}$ **do**
6:    $s^l(t) = H(mem(t-1) - \theta^*) \cdot \theta^*$, where $s^l(t)$ is the spike output scaled by $\theta^*$, and $H(x)$ is the Heaviside step function.
7:    $mem(t) = mem(t-1) - s^l(t)$
8: **return** $s^l$

---

(MAC) operations during ANN inference. It is important to note that the MatMul layers receive spike trains over $L_n$ timesteps from the preceding IF layer during SNN inference. The binary spike at each timestep is represented by $\{0, 1\} \cdot \theta^*$, where the threshold $\theta^* = \frac{\theta_n}{L_n}$. We would like to point out that the quantity obtained by scaling a spike of unit magnitude with the threshold is referred to as *postsynaptic potential* in the literature (Bu et al., 2023). Therefore, all computations in MatMul layers are **unrolled** over time and performed on binary input. The former MAC operations are replaced by repeated additions carried out on the spike train.

### 3.3 Demonstration of Correctness of PASC Algorithm

We characterize the correctness of PASC algorithm by defining the notion of *mathematical equivalence* (Definition 3.2), which captures whether two models produce the same *classification maps* (Definition 3.1) for every input matrix.

**Definition 3.1.** We define the ***classification map*** of a model $\mathcal{M}$ with respect to a given input $z$ as the probability distribution function $f_{\mathcal{M}}^z : \mathbf{C} \to [0, 1]$, where $\mathbf{C}$ is the set of output classes. The function $f$ essentially specifies the class-wise prediction probabilities of the model.

**Definition 3.2.** Two classification models $\mathcal{M}$ and $\mathcal{N}$ are said to be ***mathematically equivalent*** if their *classification maps* (Definition 3.1) are identical for all inputs. In other words, for all inputs $z$, $f_{\mathcal{M}}^z = f_{\mathcal{N}}^z$, i.e. $f_{\mathcal{M}}^z(c) = f_{\mathcal{N}}^z(c)$, $\forall c \in \mathbf{C}$.

In order to establish this property at the final layer, we show that we maintain the following invariant at the end of each layer $l$.

$$\sum_{i=1}^{L_n} z_i^l = z^l \tag{2}$$

Here, $z_i^l$ denotes the $i^{th}$ unrolled dimension of the output tensor computed using the PASC formulation, and $z^l$ is the output tensor in the corresponding ANN with QCFS activation. We state theorems for two classes of operations. The notations used in each theorem are specified in Table 1.

1. **MatMul operation (either convolution or fully-connected) + Batch Normalization** (Theorem 3.3)

   **Theorem 3.3.** *Consider a layer with quantization step $L$, performing a MatMul operation followed by Batch Normalization, i.e. $z^{l+1} = \gamma \cdot \frac{(z^l * W) + b - \mu}{\sqrt{\sigma^2 + \epsilon}} + \beta$. Assume that $z^l = \sum_{i=1}^L z_i^l$. Then, if $z_i^{l+1} = \gamma \cdot \frac{(z_i^l * W + b' - \mu')}{\sqrt{\sigma^2 + \epsilon}} + \beta'$, $\forall i \in \{1, 2, \dots L\}$, where $b' = \frac{b}{L}$, $\mu' = \frac{\mu}{L}$ and $\beta' = \frac{\beta}{L}$, then $z^{l+1} = \sum_{i=1}^L z_i^{l+1}$.*

2. **Integrate-and-Fire operation** (Theorem 3.4)

   **Theorem 3.4.** *Let $z^l$ be the input to a QCFS activation layer of an ANN with threshold $\theta_n$. Suppose that the quantization step is $L_{n-1}$ for the input. Consider a set of tensors $z_i^l$, for $i \in \{1, 2, \dots L_{n-1}\}$ such that $\sum_i z_i^l = z^l$. Let $X$ be a matrix formed by stacking $z_i^l$, i.e. $X[i] = z_i^l$, $\forall i \in \{1, 2, \dots L_{n-1}\}$. Then, **(a)** $\widehat{h}(z^l) = \lambda^l \operatorname{clip}\left(\frac{1}{L_n} \left\lfloor \frac{z^l L_n}{\lambda^l} + \frac{1}{2} \right\rfloor, 0, 1\right) = \sum_j \widetilde{h}(X)[j], \forall j \in \{1, 2, \dots L_n\}$, where $\widetilde{h}$ represents the IF operation detailed in Algorithm 1, and $L_n$ is the quantization step for the output. Moreover, **(b)** $\widetilde{h}(X)[j]$ contains values only from the set $\{0, \frac{\theta_n}{L_n}\}$.*

Both of these theorems are inductive in nature, so we also state Theorem 3.5 to show that Invariant 2 holds immediately after the input layer.

**Theorem 3.5.** *Consider a model with quantization step $L_n$ and threshold $\theta_n$ for the input layer. For a given input $z$, let the output of an ANN after the first QCFS activation layer be $z^l$ and the corresponding output in the PASC formulation be $s^l$. Then, **(a)** $\sum_{i=1}^{L_n} s^l(i) = z^l$. Moreover, **(b)** $s^l$ contains values only from the set $\{0, \frac{\theta_n}{L_n}\}$.*

We now state the overall equivalence between an SNN with PASC formulation and the corresponding ANN with QCFS activation in Theorem 3.6.

**Theorem 3.6.** *Let $\mathcal{M}$ be the PASC formulation of any ANN model $\mathcal{N}$ with QCFS activation. Then, $\mathcal{M}$ is **mathematically equivalent** to $\mathcal{N}$ (Definition 3.2).*

The proofs for the aforementioned theorems are presented in Appendix A.1.

---

**Algorithm 3** Computing the AL Metric

---

1: **Input:** The output of the QCFS activation layer of an ANN with quantization step $L$, for a batch of **I** training images. We show results for **I** = 3000.
2: Convert the input to a histogram **H**, storing the number of elements belonging to each level of QCFS activation with quantization step $L$.
3: Compute the Skewness $g$ (Equation 4), the Van Der Eijk's Agreement $A$ (refer Section 3.4.1) , and the kurtosis $K$ (Equation 5), using the histogram **H**.
4: Calculate the AL metric $M = A \times (g^2 + 1) \times K$.

---

**Algorithm 4** Comprehensive Training Methodology using Adaptive Layerwise QCFS Activation

---

1: Train a model with $L = L_\alpha$, on the input dataset until it reaches convergence. This is a standard training iteration with uniform $L_\alpha$ across all the QCFS layers. A pre-trained model can be used, if available off-the-shelf, for the subsequent steps.
2: Obtain the AL metric using Algorithm 3 for this training iteration, and partition the layers into clusters (refer Section 3.4.5). At this stage, the user is free to choose the number of clusters $\chi$ and the value of $L$ for each cluster, depending on the desired accuracy-efficiency trade-off.
3: Re-train the model, with uniform $L = L_\alpha$, for a fraction $p \approx \frac{2}{3}$ of the total number of epochs used in the initial training step.
4: Update the value of $L_n$ for each QCFS layer and train incrementally until convergence.

---

### 3.4   AL: Adaptive Layerwise QCFS Activation

In this section, we describe the proposed Adaptive Layerwise QCFS activation methodology (referred to as AL) to determine the optimal quantization step for every layer. The AL algorithm consists of the following steps.

1. Computing the AL metric, as presented in Algorithm 3.

2. Using the metric to train a new model, with adaptive layerwise quantization step, as detailed in Algorithm 4.

The AL metric is a statistical indicator of the frequency distribution of various quantization levels in the QCFS layers. We use this metric to ascertain the optimal $L_n$ per layer required to approximate the original distribution. The proposed metric is a combination of three individual statistical measures, each of which is explained in the following sections.

### 3.4.1   Van Der Eijk's Agreement

We use Van Der Eijk's agreement (Eijk, 2001) to determine how many bins in the histogram **H** have a sufficient frequency. It is specified by

$$A = 1 - \frac{S - 1}{K - 1}, \tag{3}$$

where $A$ indicates the Van Der Eijk's agreement, $S$ is the number of non-empty categories, and $K$ is the total number of categories. We employ a threshold $\alpha \in (0, 1)$ to ascertain if a bin is non-empty, as detailed below. If the frequency of the $i^{th}$ bin, $W_{b_i}$, satisfies $W_{b_i} \geq \alpha \cdot \sum_j W_{b_j}$, we call the bin non-empty; otherwise, we call it empty. We show the effect of $\alpha$ on the final metric in Appendix A.2. The insights we draw from $A$ are twofold:

1. If $A \approx 1$, then there are few bins which are full. This is indicative of a unimodal distribution, which can be approximated well with lower precision. Such a layer is amenable to a reduction in quantization step.

2. If $A \approx 0$, then there are many bins which are full. This means that the activation requires higher precision, and hence would not support a reduction in the quantization step $L$.

### 3.4.2 Skewness

Skewness is defined as the deviation of the given distribution from a normal distribution (Joanes & Gill, 1998). It is given by

$$g = \frac{m_3}{s^3} = \frac{\frac{1}{n}\sum_{i=1}^{n}(x_i - \bar{x})^3}{\left(\frac{1}{n-1}\sum_{i=1}^{n}(x_i - \bar{x})^2\right)^{\frac{3}{2}}}, \tag{4}$$

where $n$ is the number of samples, $m_3$ is the third moment, $s$ is the sample standard deviation, and $x_i$ are the samples whose mean is $\bar{x}$. We interpret the value of skewness, as explained below.

- A larger positive skewness indicates that the peak of the given distribution is close to zero. Hence, the quantization step can be reduced without impacting the peak of the distribution.

- A smaller positive skewness indicates that the given distribution more closely approximates a normal distribution. This renders a reduction in the quantization step infeasible.

We use the metric $g^2 + 1$, where $g$ is as defined in Equation 4, as adopted in prior works (Tarbă et al., 2022).

### 3.4.3 Kurtosis

Kurtosis $K$ is formulated as

$$K = \frac{k_4}{k_2^2} = \frac{(n+1)n}{(n-1)(n-2)(n-3)} \cdot \frac{\sum_{i=1}^{n}(x_i - \bar{x})^4}{k_2^2}, \tag{5}$$

where $k_4$ is the fourth moment, $k_2$ is the second moment, $n$ is the number of samples and $x_i$ are the samples whose mean is $\bar{x}$ (Joanes & Gill, 1998). We interpret kurtosis in the following manner.

- A distribution with a large (positive) kurtosis indicates a sharp and narrow peak, which makes it possible to reduce the quantization step.

- Conversely, a low kurtosis indicates a distribution with a broader peak, which warrants more quantization levels (or a larger quantization step).

### 3.4.4 Final Metric

The final AL metric, $M$, is computed by taking the product of all the three aforementioned statistical measures, as specified below

$$M = A \times (g^2 + 1) \times K. \tag{6}$$

A higher value of $M$ indicates that a greater reduction in the quantization step is possible, and vice versa.

### 3.4.5 Adaptive Layerwise QCFS Activation based Training Methodology

We estimate the AL metric for all the QCFS layers of a pretrained ANN. We use 1-dimensional clustering (Grønlund et al., 2018) to divide the layers into $\chi$ clusters based on the AL metric values, where $\chi$ is a user-defined parameter. The goal is to group QCFS layers with similar AL metric values and assign a suitable quantization step. We subsequently retrain the source ANN from the beginning. This approach is inspired by the Lottery Ticket Hypothesis (Frankle & Carbin, 2019); the difference being that our subnetwork is not obtained by pruning, but rather by activation quantization. We employ a higher quantization step for the initial epochs to maximize the effect of the learning rate scheduler (Loshchilov & Hutter, 2016). We then configure the quantization step per layer and incrementally fine-tune the model until training converges, as outlined in Algorithm 4.

Table 2: Accuracy comparison between SNNs, inferred using PASCAL, and their source ANNs across various models and datasets.

| Dataset | Model | Values of Quantization Step $L$ across Layers | ANN Accuracy | SNN Accuracy |
|---|---|---|---|---|
| CIFAR-10 | VGG-16 | [4,4,4,4,4,4,4,4,4,4,4,4,4,4,4] | 95.66 % | 95.61 % |
| CIFAR-10 | VGG-16 | [8,8,8,8,8,8,8,8,8,8,8,8,8,8,8] | 95.79 % | 95.82% |
| CIFAR-10 | VGG-16 | [4,4,4,4,1,1,1,1,1,1,4,4,4,4,4] | 93.57 % | 93.62 % |
| CIFAR-100 | VGG-16 | [4,4,4,4,4,4,4,4,4,4,4,4,4,4,4] | 76.28 % | 76.06 % |
| CIFAR-100 | VGG-16 | [8,8,8,8,8,8,8,8,8,8,8,8,8,8,8] | 77.01 % | 76.77 % |
| ImageNet | VGG-16 | [16,16,16,16,16,16,16,16,16,16,16,16,16,16,16] | 74.16 % | 74.22 % |
| ImageNet | ResNet-34 | [8,8,8,8,8,8,8,8,8,8,8,8,8,8,8,8] | 74.182% | 74.31% |
| CIFAR-10 | ResNet-18 | [8,8,8,8,8,8,8,8,8,8,8,8,8,8,8,8] | 96.51 % | 96.51 % |
| CIFAR-100 | ResNet-18 | [8,8,8,8,8,8,8,8,8,8,8,8,8,8,8,8] | 78.43 % | 78.45% |
| CIFAR-100 | ResNet-18 | [4,2,4,1,4,2,2,1,4,2,2,1,4,1,1,1,4] | 76.38% | 76.19 % |

## 4 Results

### 4.1 PASCAL: Accuracy Results

Table 2 shows the classification accuracy of SNNs, obtained using the PASCAL methodology, compared to ANNs with QCFS Activation. Our method mirrors the accuracy of the source ANN for both uniform and non-uniform values of $L$, notwithstanding minor differences arising due to floating-point imprecision. We report competitive accuracy for SNNs inferred using the PASCAL formulation across model architectures (VGG-16, ResNet-18, and ResNet-34) and datasets (CIFAR-10, CIFAR-100, and ImageNet).

### 4.2 PASCAL: Energy Efficiency Results

#### 4.2.1 Energy of PASCAL-SNN with respect to vanilla SNN

SNNs gain their energy efficiency as a result of replacing compute-intensive Multiply-and-Accumulate (MAC) operations by more energy-efficient Accumulate (AC) operations. The energy consumed by the entire network, $E_{SNN}$, can be split into two parts as follows.

$$E_{SNN} = E_{AC} + E_{IF} \tag{7}$$

Unlike previous works (Rathi & Roy, 2021), we also take into account the energy consumed by the Integrate-and-Fire layer ($E_{IF}$). Similar to Equation 7, we express the total energy consumed by the corresponding SNN trained using the PASCAL approach as shown below.

$$E_{PASC\_SNN} = E_{PASC\_AC} + E_{PASC\_IF} \tag{8}$$

We note that the energy consumed by AC operations does not change in PASCAL compared to the vanilla SNN. This is because for a given quantization step $L$, both networks have a MatMul layer that is unrolled across $T = L$ timesteps. Therefore, we have $E_{PASC\_AC} = E_{AC}$. The only difference lies in the IF layer, which is more complex in PASCAL. Consider a network with uniform quantization step $L$. According to Algorithm 1, the number of soft reset operations is $3L - 1$, while the number of spike accumulation operations (i.e., addition and subtraction of the spike counter) is $2L - 1$. In total, each IF layer in PASC framework performs $(5L - 2)$ operations, yielding $E_{PASC\_IF} \approx (5L - 2) \cdot E_{IF}$. We quantify the excess energy consumed by the PASC SNN by comparing it with the energy consumed by the vanilla SNN. Specifically, we define the *per-layer energy ratio* $r_E^l$ as

$$r_E^l = \frac{E_{PASC\_SNN}^l}{E_{SNN}^l} = \frac{E_{AC}^l + (5L-2) \cdot E_{IF}^l}{E_{AC}^l + L \cdot E_{IF}^l}. \tag{9}$$

We now analytically compute the ratio from Equation 9 for both convolutional and fully connected layers. To facilitate this, we introduce an auxiliary ratio $r'$ that represents the number of operations in an IF layer relative to those in the preceding MatMul layer. This allows us to express $r_E^l$ in terms of $r'$ as

$$r_E^l = \frac{1 + (5L-2) \cdot r'}{1 + L \cdot r'}. \tag{10}$$

Finally, we proceed to compute $r'$ layerwise separately for both the convolution and fully connected layers, using notation outlined in Table A.8

**Convolution Layer**

We assume a constant quantization step $L$ for further calculations in this section. Each IF layer performs $L \times C_0 \times H_0 \times W_0$ accumulation operations. Therefore, we compute the ratio between the number of operations in the IF layer to the number of operations in the previous convolution layer as

$$r'_{conv} = \frac{E_{IF}^l}{E_{AC}^l} = \frac{\#OP_{IF}^l}{\#OP_{Conv}^l} \approx \frac{1}{C_i \cdot K_h \cdot K_w \cdot SpikeRate_l}, \tag{11}$$

where $SpikeRate_l$ is the total spikes in layer $l$ across all timesteps averaged over the number of neurons in the layer.

**Fully Connected (FC) Layer**

Following a similar reasoning to the previous section, we can deduce $r'$ for the fully connected layer as

$$r'_{FC} = \frac{E_{IF}^l}{E_{AC}^l} = \frac{\#OP_{IF}^l}{\#OP_{FC}^l} \approx \frac{1}{C_i \cdot SpikeRate_l}. \tag{12}$$

We use the above equations defining $r'$ and $r_E^l$ to find the normalized number of inference timesteps $T_{norm}$ in the PASC framework as

$$T_{norm} = \left( \frac{1}{|\mathbf{L_{tot}}|} \cdot \sum_l r_E^l \right) \cdot T, \tag{13}$$

where $\mathbf{L_{tot}}$ is the total number of layers and $T$ is the number of time-steps of the corresponding vanilla SNN. For this computation, we assume that the $SpikeRate_l = 0.75$ for all layers. In addition, we also define the overall energy ratio as

$$r_E = \frac{E_{PASC\_SNN}^{tot}}{E_{SNN}^{tot}}, \tag{14}$$

which characterizes the overall energy cost of our method with respect to the traditional SNN. We show in Appendix A.10 that the ratio $r_E$ is as low as 1.001 in practice, indicating that increasing the number of IF operations does not adversely affect the overall energy consumption. This shows the ability of the proposed PASC framework to provide lossless ANN-SNN conversion for negligible increase in energy consumption.

### 4.2.2 Energy comparison of vanilla SNN with ANN

Having established that PASCAL preserves the energy benefits of vanilla SNNs, we now analytically compute the energy efficiency of an SNN relative to its source ANN. We estimate this energy efficiency by comparing the number of operations between the source ANN ($\#OP_{ANN}$) and the converted SNN ($\#OP_{SNN}$), following the approach proposed by Rathi & Roy and formulated as

$$\#OP_{SNN} = SpikeRate_l \times \#OP_{ANN}. \tag{15}$$

Table 3: Energy comparison between ANN and SNN based on the method proposed by Rathi & Roy. The FP32 MAC (AC) energy is estimated as 4.6 $pJ$ (0.9 $pJ$) and the INT8 MAC(AC) energy is estimated as 0.23 $pJ$ (0.03 $pJ$) (Horowitz, 2014). The spike rate per layer for various models is shown in Appendix A.9.

| Architecture (quantization step) | Dataset | Normalized $\#OP_{ANN}(a)$ | Normalized $\#OP_{SNN}(b)$ [1] | $\frac{\#OP_{SNN\_Layer1}}{\#OP_{Total}}(c)$ | $\frac{Energy_{ANN}}{Energy_{SNN}}(FP32)$ $\left(\frac{a*4.6}{c*4.6+(1-c)*b*0.9}\right)$ | $\frac{Energy_{ANN}}{Energy_{SNN}}(INT8)$ $\left(\frac{a*0.23}{c*0.23+(1-c)*b*0.03}\right)$ |
|---|---|---|---|---|---|---|
| VGG-16 (L=4) | CIFAR-10 | 1.0 | 0.66 | 0.005 (Appendix A.5) | 7.49 | 11.03 |
| VGG-16 (L=4) | CIFAR-100 | 1.0 | 0.62 | 0.005 (Appendix A.5) | 7.96 | 11.70 |
| VGG-16 (L=16) | ImageNet | 1.0 | 0.73 | 0.006 (Appendix A.6) | 6.76 | 9.94 |
| ResNet-18 (L=8) | CIFAR-10 | 1.0 | 0.86 | 0.008 (Appendix A.7) | 5.72 | 8.38 |
| ResNet-18 (L=8) | CIFAR-100 | 1.0 | 0.91 | 0.008 (Appendix A.7) | 5.42 | 7.95 |

[1] Average $SpikeRate_l$ across layers, is computed as $\frac{\sum_{l=1}^{|\mathbf{L_{tot}}|} SpikeRate_l}{|\mathbf{L_{tot}}|}$, where $|\mathbf{L_{tot}}|$ is the total number of layers.

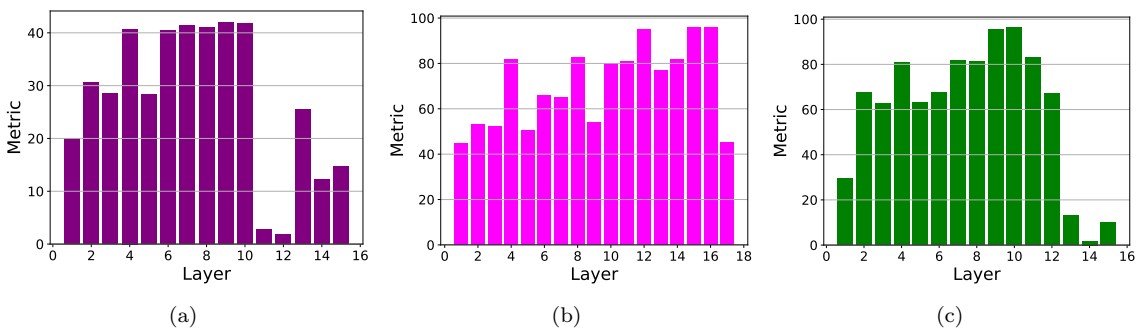

      (a)             (b)             (c)

Figure 1: AL metric for different models and datasets, with the threshold $\alpha = \frac{1}{2\cdot|\mathbf{L_{tot}}|}$ for Agreement, where $\mathbf{L_{tot}}$ is the number of layers. AL metric for (a) VGG-16 ($L = 4$) on CIFAR-10, (b) ResNet-18 ($L = 8$) on CIFAR-10, and (c) VGG-16 ($L = 8$) on CIFAR-100.

Table 4: SNN accuracy and effective timesteps obtained using the Adaptive Layerwise (AL) activation method for different models.

| AL Metric | Model | Dataset | $\chi$ | L | Layerwise L | $T_{eff}$ | Accuracy (%) |
|---|---|---|---|---|---|---|---|
| Figure 1(a) | VGG-16 | CIFAR-10 | 3 | 1,2,4 | [2,1,2,1,2,1,1, 1,1,1,4,4,2,4,4] | 2.13 | 93.73 |
| Figure 1(b) | ResNet-18 | CIFAR-10 | 2 | 1,4 | [4,4,4,1,4,4,4,1,4,1,1,1,1,1,1,1,4] | 3.14 | 95.34 |
| Figure 1(c) | VGG-16 | CIFAR-100 | 3 | 1,4,8 | [8,4,4,1,4,4,1,1 ,1,1,1,4,8,8,8] | 4.54 | 74.07 |
| | VGG-16 | ImageNet | 3 | 16,8,4 | [16,16,16,8,8,8,8,8,4,4,4,4,4,4,4] | 9.92 | 71.188 |
| | VGG-16 | ImageNet | 2 | 16,8 | [16, 16, 16, 8, 16, 8, 8, 8, 8, 8, 8, 8, 8, 8, 8] | 12.40 | 73.98 |
| | ResNet-34 | ImageNet | 2 | 16,8 | [16, 16, 16, 8, 16, 8, 16, 8, 16, 8, 16, 8, 16, 8, 16, 8, 16, 8, 16, 8, 16, 8, 16, 8, 16, 8, 8, 8, 8, 8, 8] | 12.94 | 75.78 |

Table 3 shows that converted SNNs are up to 7× more energy efficient than the corresponding source ANNs. The energy results account for the fact that the first SNN layer alone performs MAC operations since it receives input image with real-valued pixel intensities. The formulae used to compute the ANN and SNN operations are detailed in Appendix A.4.

Table 5: SNN accuracy and inference timesteps comparison with QCFS and SEENN-1.

| PASC formulation with uniform Quantization Step | | | | | PASC with Adaptive Layerwise (AL) activation | | | | |
|---|---|---|---|---|---|---|---|---|---|
| Model | Dataset | Method | T | Acc. | Model | Dataset | Method | T | Acc. |
| VGG-16 | CIFAR-10 | QCFS | 16 | 93.95% | ResNet-18 | CIFAR-10 | QCFS | 8 | 94.82% |
|  |  | **PASC** ($T_{norm}$) | 4.21 | 95.61% |  |  | SEENN-1 | 2.01 | 95.08% |
|  | CIFAR-100 | QCFS | 16 | 72.80% |  |  | **PASCAL** ($T_{eff}$) | 3.14 | 95.34% |
|  |  | **PASC** ($T_{norm}$) | 4.21 | 76.06% |  | CIFAR-100 | SEENN-1 | 6.19 | 65.48% |
| ResNet-34 | ImageNet | QCFS | 64 | 72.35% |  |  | **PASCAL** ($T_{eff}$) | 2.63 | 76.19 % |
|  |  | QCFS | 256 | 74.22% | VGG-16 | CIFAR-100 | QCFS | 16 | 72.80% |
|  |  | SEENN-1 | 29.53 | 71.84% |  |  | **PASCAL** ($T_{eff}$) | 4.54 | 74.07% |
|  |  | **PASC** ($T_{norm}$) | 8.17 | 74.31% | VGG-16 | ImageNet | **PASCAL** ($T_{eff}$) | 12.40 | 73.98 % |
| VGG-16 | ImageNet | QCFS | 1024 | 74.32% | ResNet-34 | ImageNet | **PASCAL** ($T_{eff}$) | 12.94 | 75.78 % |
|  |  | **PASC** ($T_{norm}$) | 18.37 | 74.22% |  |  |  |  |  |

Table 6: SNN accuracy and inference timesteps comparison with other state-of-the-art works.

| Prior Work | Dataset | Model | Accuracy of Prior Work(%) | Accuracy of PASC(%) |
|---|---|---|---|---|
| Hao et al. (2023) | CIFAR-100 | VGG-16 | 76.77 (T=16) | 76.77 ($T_{norm} = 8.79$) |
| Hao et al. (2023) | ImageNet | ResNet-34 | 73.93 (T=32) | 74.31 ($T_{norm} = 8.17$) |
| Li et al. (2023a) | CIFAR-10 | ResNet-18 | 94.27 (T=11.51) | 96.51 ($T_{norm} = 8.22$) |
| Li et al. (2023a) | ImageNet | VGG-16 | 73.30 (T=49.54) | 74.22 ($T_{norm} = 18.37$) |
| Wu et al. (2024) | ImageNet | ResNet34 | 71.66 (T=16) | 74.31 ($T_{norm} = 16.67$) |

## 4.3 Adaptive Layerwise (AL) Activation Results

Figure 1 shows the layerwise AL metric (Equation 6) for different models and datasets. We perform clustering with various values of $\chi$, and compute the quantization step per layer. We also evaluate the effective inference timesteps, $T_{eff}$, which is specified by

$$T_{eff} = \frac{1}{|\mathbf{L_{tot}}|} \cdot \sum_l r_E^l \cdot T^l, \tag{16}$$

where $\mathbf{L_{tot}}$ is the total number of layers and $T^l$ is the number of time-steps of the corresponding vanilla SNN for the same layer. Table 4 shows that adapting the quantization step in a layerwise manner achieves competitive accuracy using fewer effective timesteps, thereby improving temporal efficiency. For instance, ResNet-18 provides 95.34% accuracy on CIFAR-10 using $T_{eff} \approx 3.14$, which is comparable to the 96.51% accuracy achieved with a uniform $T_{norm} = 8.22$ (see Table 6).

## 4.4 Comparison with the State-of-the-Art

We compare our results, presented in Table 5, with those of the QCFS framework (Bu et al., 2023) and the SEENN-1 method (Li et al., 2023b). In addition to these baselines, we also benchmark our approach against various other state-of-the-art works listed in Table 6. Our results indicate that PASCAL consistently yields SNNs that provide competitive accuracy using fewer timesteps across models and datasets, outperforming the QCFS framework (Bu et al., 2023). We report comparable performance to the SEENN-1 method (Li et al., 2023b) on CIFAR-10. However, we demonstrate significant performance improvements, both in terms of accuracy and effective timesteps, for CIFAR-100 and ImageNet. This trend is also observed when comparing

our method with more recent works in Table 6: PASCAL remains competitive on smaller datasets and achieves superior results on larger ones. The efficacy of PASCAL improves with dataset complexity since it relies on mathematically equivalent ANN-SNN conversion.

## 5 Conclusion

In this paper, we proposed PASCAL, which is a precise and temporally efficient ANN-SNN conversion method. We showed that the proposed methodology enables ANNs to be converted into SNNs with a guarantee of mathematical equivalence, leading to near-zero conversion loss. We formally proved that the converted SNN performs the same computations, unrolled over timesteps, as the source ANN with QCFS activation. We demonstrated the energy efficiency of the proposed approach using an analytical model. In addition, we also proposed an Adaptive Layerwise (AL) activation methodology to further improve temporal efficiency during SNN inference. We presented an algorithmic approach to choose an optimal precision for each layer, which reduces the effective number of inference timesteps while providing competitive accuracy.

## 6 Acknowledgements

This research was supported in part by the RISC-V Knowledge Centre of Excellence (RKCoE), sponsored by the Ministry of Electronics and Information Technology (MeitY). We would like to acknowledge the Robert Bosch Centre for Data Science and AI (RBCDSAI) and the Centre for Development of Advanced Computing (C-DAC) for providing us GPU computing resources. Finally, we would like to acknowledge the work of Varun Manjunath (Electrical Engineering, IIT Madras, Class of 2025) for benchmarking our work on neuromorphic architectural simulators.

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

# A Appendix

## A.1 Proof for equivalence between the PASC formulation and an ANN with QCFS activation

**Proof for Theorem** 3.5.

For the input layer, the MatMul operation performed in the PASC formulation is the same as the corresponding ANN with QCFS activation. Let the output after the MatMul Operation be $X$. Consider a single element $x \in X$. In the case of ANN, the output after QCFS is

$$\widehat{h}(x) = \theta_n \, \text{clip}\left(\frac{1}{L_n}\left\lfloor \frac{XL_n}{\theta_n} + \varphi \right\rfloor, 0, 1\right) = c \cdot \frac{\theta_n}{L_n}. \tag{17}$$

Upon further simplification, we obtain

$$c = \begin{cases} k & \text{if } 0 < k < L_n, \text{ and} \\ & (k - \frac{1}{2})\frac{\theta_n}{L_n} \leq x < (k + \frac{1}{2})\frac{\theta_n}{L_n} \\ L_n & \text{if } x \geq (L_n - \frac{1}{2})\frac{\theta_n}{L_n} \\ 0 & \text{if } x < \frac{\theta_n}{2 \cdot L_n} \end{cases} \tag{18}$$

Algorithm 2 specifies the Integrate-and-Fire algorithm for the input layer. It also first generates $\widehat{h}(X) = z^l$ (refer to Line 3). Subsequently, it generates the final spike train $s^l$. We denote the spike train for a given input $x \in X$ by $s_x^l$. Note that $z^l$ only consists of integral multiples of $\theta^* = \frac{\theta_n}{L_n}$. Therefore, after repeated application of the Heaviside step function on $x$, and stacking the output over time, we get

$$s_x^l(i) = \begin{cases} \frac{\theta_n}{L_n} & \text{if } i \leq c \\ 0 & \text{otherwise} \end{cases} \tag{19}$$

Now, $\sum_{i=1}^{L_n} s_x^l(i) = c \cdot \frac{\theta_n}{L_n} = \widehat{h}(x)$, $\forall x \in X$. Therefore, $\sum_{i=1}^{L_n} s^l(i) = z^l$, proving part (a). Using Equation 19, we can see that $s_x^l$ contains values from the set $\{0, \frac{\theta_n}{L_n}\}$ for all values of $x \in X$. This proves part (b) of the theorem. $\square$

**Proof for Theorem** 3.4. We begin with the remark that $\theta_n = \lambda^l$ after the training is complete. Consider a single element $x \in z^l$ and the corresponding elements $x_i \in z_i^l, \forall i \in \{1, 2, \cdots L_{n-1}\}$. In an ANN with QCFS activation (see Figure 2),

$$\widehat{h}(x) = \begin{cases} k \cdot \frac{\theta_n}{L_n} & \text{if } 0 < k < L_n, \text{ and} \\ & (k - \frac{1}{2})\frac{\theta_n}{L_n} \leq x < (k + \frac{1}{2})\frac{\theta_n}{L_n} \\ \theta_n & \text{if } x \geq (L_n - \frac{1}{2})\frac{\theta_n}{L_n} \\ 0 & \text{if } x < \frac{\theta_n}{2 \cdot L_n} \end{cases} \tag{20}$$

Table 7: Mapping between the value of $s$ at the end of stage 2, and the final number of spikes at the end of stage 3.

| Value of $s$ | Number of Spikes |
|---|---|
| $2L_n - 1 \geq s \geq L_n$ | $L_n$ |
| $L_n > s > 0$ | $s$ |
| $0 \geq s \geq -(L_n - 1)$ | $0$ |

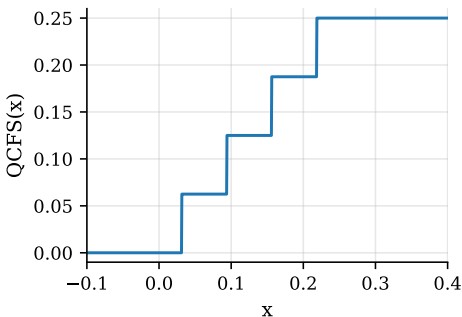

Figure 2: A sample QCFS activation layer with $L = 4$ and $\theta_n = 0.25$.

After the first stage of Algorithm 1, let the value of $s$ be $k_1 \cdot \frac{\theta_n}{L_n}$. If there was a spike in the last timestep, then we obtain an inequality of the form $\frac{\theta_n}{2 \cdot L_n} + \sum_i x_i - k_1 \cdot \frac{\theta_n}{L_n} \geq \frac{\theta_n}{L_n}$. Upon simplification, we get

$$\sum_{i=1}^{L_{n-1}} x_i \geq (k_1 + \frac{1}{2}) \cdot \frac{\theta_n}{L_n} \tag{21}$$

On the other hand, if there was no spike in the last timestep, we instead obtain

$$\sum_{i=1}^{L_{n-1}} x_i < (k_1 + \frac{1}{2}) \cdot \frac{\theta_n}{L_n} \tag{22}$$

Although we have a one-directional inequality in either case, we nevertheless need more processing to obtain the second direction of the inequality to choose the correct level. We consider the aforementioned two cases.
**Case 1.** Let Inequality 21 be the final inequality after stage 1. Inhibitory spikes are not possible in this case. The QCFS function clips the activation if the accumulated input $\sum_i x_i$ is greater than $(L_n - \frac{1}{2}) \cdot \frac{\theta_n}{L_n}$. Therefore, the IF step needs to be done without input at least $T_1 = L_n - 1$ times to ensure that the correct level is obtained for all inputs. The bound arises because the minimum value of $k_1 = 0$. At the end of stage 2, let the value of $s$ be $k_2 \cdot \frac{\theta_n}{L_n}$. Since an inhibitory spike is not possible, we get

$$\frac{\theta_n}{2 \cdot L_n} + \sum_i x_i - k_2 \cdot \frac{\theta_n}{L_n} \geq 0 \tag{23}$$

Also, if $k_2 < L_n$, then we also have the following inequality.

$$\frac{\theta_n}{2 \cdot L_n} + \sum_i x_i - k_2 \cdot \frac{\theta_n}{L_n} < \frac{\theta_n}{L_n} \tag{24}$$

Otherwise, we only have the inequality specified in Equation 23. In effect,

$$s = \begin{cases} k \cdot \frac{\theta_n}{L_n}, k < L_n & \text{if } (k - \frac{1}{2})\frac{\theta_n}{L_n} \\ & \leq \sum_i x_i < (k + \frac{1}{2})\frac{\theta_n}{L_n} \\ k \cdot \frac{\theta_n}{L_n}, k \geq L_n & \text{if } \sum_i x_i \geq (L_n - \frac{1}{2})\frac{\theta_n}{L_n} \end{cases} \tag{25}$$

**Case 2.** If Inequality 22 is the final inequality, we need **inhibitory spikes** to determine the correct level. This is different from Case 1, because we have an upper bound on the value of $\sum_{i=1}^{L_{n-1}} x_i$. In effect, any level below $k_1$ should be possible. Since $k_1 \leq L_{n-1} - 1$, this stage requires $T_2 = L_{n-1} - 1$ timesteps. Here, normal spikes are not possible. Using a similar analysis as presented in Case 1, we obtain

$$s = \begin{cases} k \cdot \frac{\theta_n}{L_n}, k > 0 & \text{if } (k - \frac{1}{2})\frac{\theta_n}{L_n} \\ & \leq \sum_i x_i < (k + \frac{1}{2})\frac{\theta_n}{L_n} \\ k \cdot \frac{\theta_n}{L_n}, k \leq 0 & \text{if } \sum_i x_i < \frac{\theta_n}{2 \cdot L_n} \end{cases} \tag{26}$$

Considering Cases 1 and 2, we run stage 2 for $max(T_1, T_2) = max(L_{n-1}, L_n) - 1$ timesteps.

Stage 3 is used to generate a final spike train from the value of $s$ obtained after stage 2. This layer is used to generate a stacked output $\widetilde{h}(X)$. As a result of this computation, we get that $\sum_j \widetilde{h}(X)[j] = min(max(s, 0), L_n)$. The final number of spikes, parametrized by the value of $s$ at the end of stage 2, is given in Table 7. Using 25 and 26, we can conclude that

$$\sum_j \widetilde{h}(x)[j] = \begin{cases} k \cdot \frac{\theta_n}{L_n} & \text{if } 0 < k < L_n, \text{ and} \\ & (k - \frac{1}{2})\frac{\theta_n}{L_n} \\ & \leq x < (k + \frac{1}{2})\frac{\theta_n}{L_n} \quad = \widehat{h}(x) \\ \theta_n & \text{if } x \geq (L_n - \frac{1}{2})\frac{\theta_n}{L_n} \\ 0 & \text{if } x < \frac{\theta_n}{2 \cdot L_n} \end{cases} \tag{27}$$

$\forall x \in X$. This proves part (a). $s^l(t)$ is the output of the Heaviside function, i.e. $\widetilde{h}(x)[j] \in \{0, \theta^* = \frac{\theta_n}{L_n}\} \ \forall j$, thus proving part (b). $\qquad\square$

**Proof for Theorem** 3.3 We begin by simplifying the right-hand side.

$$\sum_{i=1}^{L} z_i^{l+1} = \sum_{i=1}^{L} (\gamma \cdot \frac{(z_i^l * W + \frac{b}{L} - \frac{\mu}{L})}{\sqrt{\sigma^2 + \epsilon}} + \frac{\beta}{L}) \tag{28}$$

$$= \sum_{i=1}^{L} \gamma \cdot \frac{z_i^l * W}{\sqrt{\sigma^2 + \epsilon}} + \frac{1}{\sqrt{\sigma^2 + \epsilon}} \sum_{i=1}^{L} \frac{b - \mu}{L} + \sum_{i=1}^{L} \frac{\beta}{L} \tag{29}$$

$$= S_1 + S_2 + S_3 \tag{30}$$

Here,

$$S_1 = \frac{\gamma}{\sqrt{\sigma^2 + \epsilon}} \cdot (\sum_{i=1}^{L} z_i^l) * W \tag{31}$$

$$= \frac{\gamma}{\sqrt{\sigma^2 + \epsilon}} \cdot z^l * W \tag{32}$$

$$= \frac{\gamma}{\sqrt{\sigma^2 + \epsilon}} \cdot (z^{l+1} - \beta) \cdot \frac{\sqrt{\sigma^2 + \epsilon}}{\gamma} - b + \mu \tag{33}$$

$$= z^{l+1} - \beta - b + \mu \tag{34}$$

Also,

$$S_2 = \sum_{i=1}^{L} \frac{b}{L} = b - \mu \tag{35}$$

$$S_3 = \sum_{i=1}^{L} \frac{\beta}{L} = \beta \tag{36}$$

Therefore,

$$S_1 + S_2 + S_3 = z^{l+1} - \beta - b + \mu + b - \mu + \beta \tag{37}$$

$$= z^{l+1} \tag{38}$$

$\qquad\square$

**Proof for Theorem** 3.6

Consider an input $z$ to the models $\mathcal{M}$ and $\mathcal{N}$. We invoke Theorems 3.5, 3.4 and 3.3 to show that Invariant 2 holds after the final layer. In other words, after the final layer, $z^l = \sum_{i=1}^{L} z_i^l$, where $L$ is the quantization step of the final layer $l$. In an SNN, the output probabilities are derived by taking the mean over timesteps after the

final layer. Therefore, the tensor used for classification in the PASC formulation is $z^* = \frac{1}{L} \cdot \sum_{i=1}^{L} z_i^l = \frac{1}{L} \cdot z^l$. Therefore,

$$f_{\mathcal{M}}^z(c) = \frac{z^*[c]}{\sum_c z^*[c]} \tag{39}$$

$$= \frac{\frac{z^l[c]}{L}}{\frac{\sum_c z^l[c]}{L}} \tag{40}$$

$$= \frac{z^l[c]}{\sum_c z^l[c]} \tag{41}$$

$$= f_{\mathcal{N}}^z(c) \tag{42}$$

$\forall c \in \mathbf{C}$. Thus, $\mathcal{M}$ and $\mathcal{N}$ are mathematically equivalent. $\qquad\square$

## A.2  Effect of threshold of Van Der Eijk's A on the Final Metric

The effect of A is pronounced in datasets such as CIFAR-100. It primarily affects the value of the metric in the input layer. The effect can be seen in Figure 3.

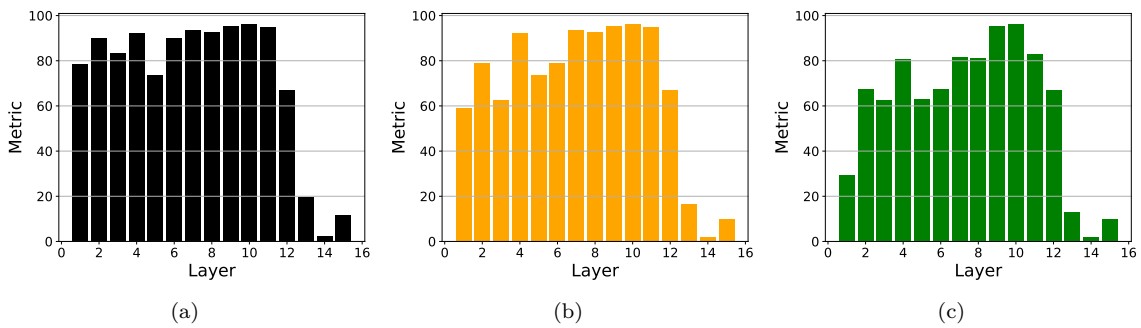

(a)        (b)        (c)

Figure 3: Layerwise AL-metric with the threshold $\alpha = \frac{k}{|\mathbf{L_{tot}}|}$ for Agreement, where $\mathbf{L_{tot}}$ is the number of layers, computed using VGG-16 for CIFAR-100. (a) $k = 1$ (b) $k = 0.75$ (c) $k = 0.5$.

## A.3  Introduction of Datasets

**CIFAR-10.** The CIFAR-10 dataset (Krizhevsky et al., 2009) consists of 60000 $32 \times 32$ images in 10 classes. There are 50000 training images and 10000 test images.

**CIFAR-100.** The CIFAR-100 dataset (Krizhevsky et al., 2009) consists of 60000 $32 \times 32$ images in 100 classes. There are 50000 training images and 10000 test images.

**ImageNet.** We use the ILSVRC 2012 dataset (Russakovsky et al., 2015), which consists of 1,281,167 training images and 50000 testing images.

## A.4  Comparison of the number of MACs and ACs between ANN and SNN

The table below shows the number of operations for both convolution and fully connected layers. Table 8 shows the notation used for this analysis. The energy cost of addition and multiplication in 45nm CMOS is shown in Table 10.

| Type of Layer | Nature of Activation | #MACs | #ACs |
|---|---|---|---|
| Fully Connected | ANN | $C_i \times C_o$ | 0 |

| Type of Layer | Nature of Activation | #MACs | #ACs |
|---|---|---|---|
| Fully Connected | QCFS Activation with L levels, inferred using PASCAL | 0 | $C_i \times C_o \times SpikeRate_l$, along with $C_o \times SpikeRate_l$ multiplies for threshold scaling |
| Convolution | ANN | $C_i \times C_o \times K_h \times K_w \times W_o \times H_o$ | 0 |
| Convolution | QCFS Activation with L levels, inferred using PASCAL | 0 | $C_i \times C_o \times K_h \times K_w \times W_o \times H_o \times SpikeRate_l$ (along with $C_o \times H_o \times W_o \times SpikeRate_l$ multiplies for threshold scaling) |

## A.5 Layerwise count of operations in the VGG-16 architecture for CIFAR-10 and CIFAR-100

| Layer | Input Size | Output Size | #MACs |
|---|---|---|---|
| Conv1_1 (Convolution) | $32 \times 32 \times 3$ | $32 \times 32 \times 64$ | $32 \times 32 \times 3 \times 3 \times 3 \times 64 = 1,769,472$ |
| Conv1_2 (Convolution) | $32 \times 32 \times 64$ | $32 \times 32 \times 64$ | $32 \times 32 \times 64 \times 3 \times 3 \times 64 = 37,748,736$ |
| MaxPool1 | $32 \times 32 \times 64$ | $16 \times 16 \times 64$ | — |
| Conv2_1 (Convolution) | $16 \times 16 \times 64$ | $16 \times 16 \times 128$ | $16 \times 16 \times 64 \times 3 \times 3 \times 128 = 18,874,368$ |
| Conv2_2 (Convolution) | $16 \times 16 \times 128$ | $16 \times 16 \times 128$ | $16 \times 16 \times 128 \times 3 \times 3 \times 128 = 37,748,736$ |
| MaxPool2 | $16 \times 16 \times 128$ | $8 \times 8 \times 128$ | — |
| Conv3_1 (Convolution) | $8 \times 8 \times 128$ | $8 \times 8 \times 256$ | $8 \times 8 \times 128 \times 3 \times 3 \times 256 = 18,874,368$ |
| Conv3_2 (Convolution) | $8 \times 8 \times 256$ | $8 \times 8 \times 256$ | $8 \times 8 \times 256 \times 3 \times 3 \times 256 = 37,748,736$ |
| Conv3_3 (Convolution) | $8 \times 8 \times 256$ | $8 \times 8 \times 256$ | $8 \times 8 \times 256 \times 3 \times 3 \times 256 = 37,748,736$ |
| MaxPool3 | $8 \times 8 \times 256$ | $4 \times 4 \times 256$ | — |
| Conv4_1 (Convolution) | $4 \times 4 \times 256$ | $4 \times 4 \times 512$ | $4 \times 4 \times 256 \times 3 \times 3 \times 512 = 18,874,368$ |
| Conv4_2 (Convolution) | $4 \times 4 \times 512$ | $4 \times 4 \times 512$ | $4 \times 4 \times 512 \times 3 \times 3 \times 512 = 37,748,736$ |
| Conv4_3 (Convolution) | $4 \times 4 \times 512$ | $4 \times 4 \times 512$ | $4 \times 4 \times 512 \times 3 \times 3 \times 512 = 37,748,736$ |
| MaxPool4 | $4 \times 4 \times 512$ | $2 \times 2 \times 512$ | — |
| Conv5_1 (Convolution) | $2 \times 2 \times 512$ | $2 \times 2 \times 512$ | $2 \times 2 \times 512 \times 3 \times 3 \times 512 = 9,437,184$ |
| Conv5_2 (Convolution) | $2 \times 2 \times 512$ | $2 \times 2 \times 512$ | $2 \times 2 \times 512 \times 3 \times 3 \times 512 = 9,437,184$ |
| Conv5_3 (Convolution) | $2 \times 2 \times 512$ | $2 \times 2 \times 512$ | $2 \times 2 \times 512 \times 3 \times 3 \times 512 = 9,437,184$ |
| MaxPool5 | $2 \times 2 \times 512$ | $1 \times 1 \times 512$ | — |
| FC1 (Fully Connected) | $1 \times 1 \times 512 = 512$ | 4096 | $512 \times 4096 = 2,097,152$ |
| FC2 (Fully Connected) | 4096 | 4096 | $4096 \times 4096 = 16,777,216$ |
| FC3 (Fully Connected) | 4096 | 100 (CIFAR-100), 10 (CIFAR-10) | **40,960** (CIFAR-10) **409,600** (CIFAR-100) |

| Layer | Input Size | Output Size | #MACs |
|---|---|---|---|
| **Total Operations** | | | **332,480,512** (CIFAR-100) **332,111,872** (CIFAR10) |

## A.6 Layerwise count of operations in the VGG-16 architecture for ImageNet

| Layer | Input Size | Output Size | #MACs |
|---|---|---|---|
| Conv1_1 (Convolution) | $224 \times 224 \times 3$ | $224 \times 224 \times 64$ | $224 \times 224 \times 3 \times 3 \times 3 \times 64 = 86,704,128$ |
| Conv1_2 (Convolution) | $224 \times 224 \times 64$ | $224 \times 224 \times 64$ | $224 \times 224 \times 64 \times 3 \times 3 \times 64 = 1,849,688,064$ |
| MaxPool1 | $224 \times 224 \times 64$ | $112 \times 112 \times 64$ | — |
| Conv2_1 (Convolution) | $112 \times 112 \times 64$ | $112 \times 112 \times 128$ | $112 \times 112 \times 64 \times 3 \times 3 \times 128 = 924,844,032$ |
| Conv2_2 (Convolution) | $112 \times 112 \times 128$ | $112 \times 112 \times 128$ | $112 \times 112 \times 128 \times 3 \times 3 \times 128 = 1,849,688,064$ |
| MaxPool2 | $112 \times 112 \times 128$ | $56 \times 56 \times 128$ | — |
| Conv3_1 (Convolution) | $56 \times 56 \times 128$ | $56 \times 56 \times 256$ | $56 \times 56 \times 128 \times 3 \times 3 \times 256 = 924,844,032$ |
| Conv3_2 (Convolution) | $56 \times 56 \times 256$ | $56 \times 56 \times 256$ | $56 \times 56 \times 256 \times 3 \times 3 \times 256 = 1,849,688,064$ |
| Conv3_3 (Convolution) | $56 \times 56 \times 256$ | $56 \times 56 \times 256$ | $56 \times 56 \times 256 \times 3 \times 3 \times 256 = 1,849,688,064$ |
| MaxPool3 | $56 \times 56 \times 256$ | $28 \times 28 \times 256$ | — |
| Conv4_1 (Convolution) | $28 \times 28 \times 256$ | $28 \times 28 \times 512$ | $28 \times 28 \times 256 \times 3 \times 3 \times 512 = 924,844,032$ |
| Conv4_2 (Convolution) | $28 \times 28 \times 512$ | $28 \times 28 \times 512$ | $28 \times 28 \times 512 \times 3 \times 3 \times 512 = 1,849,688,064$ |
| Conv4_3 (Convolution) | $28 \times 28 \times 512$ | $28 \times 28 \times 512$ | $28 \times 28 \times 512 \times 3 \times 3 \times 512 = 1,849,688,064$ |
| MaxPool4 | $28 \times 28 \times 512$ | $14 \times 14 \times 512$ | — |
| Conv5_1 (Convolution) | $14 \times 14 \times 512$ | $14 \times 14 \times 512$ | $14 \times 14 \times 512 \times 3 \times 3 \times 512 = 462,422,016$ |
| Conv5_2 (Convolution) | $14 \times 14 \times 512$ | $14 \times 14 \times 512$ | $14 \times 14 \times 512 \times 3 \times 3 \times 512 = 462,422,016$ |
| Conv5_3 (Convolution) | $14 \times 14 \times 512$ | $14 \times 14 \times 512$ | $14 \times 14 \times 512 \times 3 \times 3 \times 512 = 462,422,016$ |
| MaxPool5 | $14 \times 14 \times 512$ | $7 \times 7 \times 512$ | — |
| FC1 (Fully Connected) | $7 \times 7 \times 512 = 25088$ | 4096 | $25088 \times 4096 = 102,760,448$ |
| FC2 (Fully Connected) | 4096 | 4096 | $4096 \times 4096 = 16,777,216$ |
| FC3 (Fully Connected) | 4096 | 1000 | $4096 \times 1000 = 4,096,000$ |
| **Total Operations** | | | **15,470,264,320** |

## A.7 Layerwise count of operations in the ResNet-18 architecture for CIFAR-10 and CIFAR-100

| Layer | Input Size | Output Size | #MACs |
|---|---|---|---|
| Initial Conv | $3 \times 32 \times 32$ | $64 \times 32 \times 32$ | 1,769,472 |
| Residual Block 1.1 Conv1 | $64 \times 32 \times 32$ | $64 \times 32 \times 32$ | 37,748,736 |
| Residual Block 1.1 Conv2 | $64 \times 32 \times 32$ | $64 \times 32 \times 32$ | 37,748,736 |
| Residual Block 1.2 Conv1 | $64 \times 32 \times 32$ | $64 \times 32 \times 32$ | 37,748,736 |
| Residual Block 1.2 Conv2 | $64 \times 32 \times 32$ | $64 \times 32 \times 32$ | 37,748,736 |
| Residual Block 2.1 Conv1 | $64 \times 32 \times 32$ | $128 \times 16 \times 16$ | 18,874,368 |
| Residual Block 2.1 Conv2 | $128 \times 16 \times 16$ | $128 \times 16 \times 16$ | 9,437,184 |
| Residual Block 2.1 Shortcut | $64 \times 32 \times 32$ | $128 \times 16 \times 16$ | 2,097,152 |
| Residual Block 2.2 Conv1 | $128 \times 16 \times 16$ | $128 \times 16 \times 16$ | 9,437,184 |
| Residual Block 2.2 Conv2 | $128 \times 16 \times 16$ | $128 \times 16 \times 16$ | 9,437,184 |
| Residual Block 3.1 Conv1 | $128 \times 16 \times 16$ | $256 \times 8 \times 8$ | 4,718,592 |
| Residual Block 3.1 Conv2 | $256 \times 8 \times 8$ | $256 \times 8 \times 8$ | 2,359,296 |
| Residual Block 3.1 Shortcut | $128 \times 16 \times 16$ | $256 \times 8 \times 8$ | 524,288 |
| Residual Block 3.2 Conv1 | $256 \times 8 \times 8$ | $256 \times 8 \times 8$ | 2,359,296 |
| Residual Block 3.2 Conv2 | $256 \times 8 \times 8$ | $256 \times 8 \times 8$ | 2,359,296 |
| Residual Block 4.1 Conv1 | $256 \times 8 \times 8$ | $512 \times 4 \times 4$ | 1,179,648 |

| Layer | Input Size | Output Size | #MACs |
|---|---|---|---|
| Residual Block 4.1 Conv2 | $512 \times 4 \times 4$ | $512 \times 4 \times 4$ | 589,824 |
| Residual Block 4.1 Shortcut | $256 \times 8 \times 8$ | $512 \times 4 \times 4$ | 262,144 |
| Residual Block 4.2 Conv1 | $512 \times 4 \times 4$ | $512 \times 4 \times 4$ | 589,824 |
| Residual Block 4.2 Conv2 | $512 \times 4 \times 4$ | $512 \times 4 \times 4$ | 589,824 |
| Final FC Layer | 512 | 100 (CIFAR-100) | 51,200 (CIFAR-100) |
| | | 10 (CIFAR-10) | 5,120 (CIFAR-10) |
| **Total Operations** | | | **217,630,720** (CIFAR-100) |
| | | | **217,584,640** (CIFAR-10) |

## A.8 Training Configurations

In order to implement PASCAL, we adopt the code framework of QCFS Bu et al. (2023). We use the Stochastic Gradient Descent optimizer (Bottou, 2012) with a momentum parameter of 0.9. The initial learning rate is set to 0.1 for CIFAR-10 and ImageNet, and 0.02 for CIFAR-100. A cosine decay scheduler (Loshchilov & Hutter, 2016) is used to adjust the learning rate. We apply a $5 \times 10^{-4}$ weight decay for CIFAR datasets while applying a $1 \times 10^{-4}$ weight decay for ImageNet.

## A.9 Layerwise spike rate over the test set

We plot the spike rate over the entire test set for various model architectures and datasets in Figure 4.

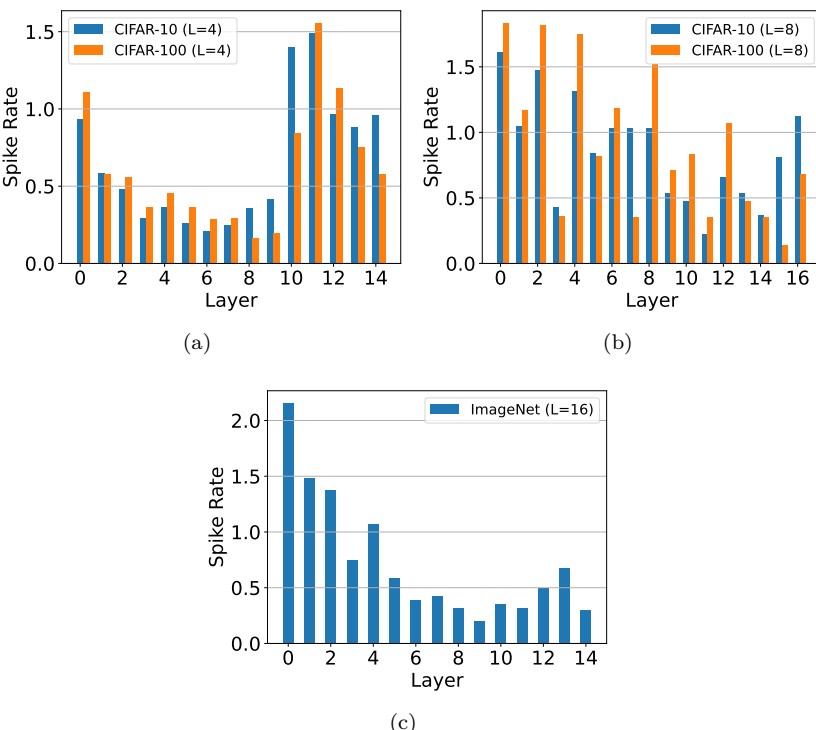

Figure 4: Layerwise Spike Rate over the test-set, (a) for $L = 4$ using VGG-16 for both CIFAR-10 and CIFAR-100, (b) for $L = 8$ using ResNet-18 for both CIFAR-10 and CIFAR-100, and (c) for $L = 16$ using VGG-16 for ImageNet.

Table 8: Notation used in computing MACs and ACs.

| Notation | Meaning |
|----------|---------|
| $C_i$ | Number of Input Channels/Features |
| $C_o$ | Number of Output Channels/Features |
| $K_h$ | Convolution Kernel Height |
| $K_w$ | Convolution Kernel Width |
| $W_o$ | Output matrix width (after convolution) |
| $H_o$ | Output matrix height (after convolution) |

Table 10: Energy cost of addition and multiplication in 45nm CMOS (Horowitz, 2014).

| | |
|---|---|
| FP ADD (32 bit) | $0.9pJ$ |
| FP MULT (32 bit) | $3.7pJ$ |
| FP MAC (32 bit) | $(0.9 + 3.7)$ $= 4.6pJ$ |
| INT8 ADD | $0.03pJ$ |
| INT8 MUL | $0.2pJ$ |
| INT8 MAC | $(0.03 + 0.2)$ $= 0.23pJ$ |

## A.10  Hardware Quantification

### A.10.1  Energy

We implemented a custom SNN accelerator in-house to quantify the energy consumed per inference with and without the PASC IF layer. The accelerator is a modified version of the LoAS accelerator (Yin et al., 2024), enhanced to support the PASC IF layer. We report the energy metrics for the VGG-16 SNN, over the CIFAR-10 test set.

Table 14: Energy metrics of VGG-16 SNN with and without PASC IF.

| Type of IF | Total Inference Energy | Energy Contributed by the IF Layer |
|------------|------------------------|-------------------------------------|
| Default IF (T = 4) | 445.47 mJ | 0.08 mJ |
| PASC IF (T = 4) | 446.02 mJ | 0.63 mJ |

Our results show that the VGG-16 SNN with PASC IF neurons consumes only **0.1%** more energy compared to the default IF implementation.

In an effort to further validate the energy efficiency results, we re-implemented an ANN accelerator SparTen in addition to the aforementioned SNN accelerator LoAS Yin et al. (2024). Both the accelerators were designed to support INT8 quantization for the weights. In addition, SparTen was enhanced to use the QCFS function to implement quantized activation. We synthesized the Register Transfer Level (Verilog RTL) design of the accelerators using the Synopsys Design Compiler on 40 nm CMOS technology to determine the operating frequency (560 MHz) and power consumption. We then developed a cycle-accurate simulator of the respective accelerators to estimate inference latency and energy. Table 15 summarizes the inference energy of VGG-16 ANN and SNN, obtained empirically from our cycle-accurate simulators, over the CIFAR-10 test set.

### A.10.2  Latency

We would like to emphasize that the proposed PASC IF neuron adds only a minor overhead to inference latency and energy cost, since the end-to-end latency and energy consumption of the network are dominated by the MatMul layers, which incur more operations than the IF layers, as discussed in Section 4.2. We quantify the overhead by implementing the PASC IF neuron on the LoAS accelerator Yin et al. (2024), which employs a fully temporal-parallel dataflow architecture designed to minimize both data movement across

Table 15: Comparison of inference energy consumption across different networks.

| Type of Network | Total Inference Energy (mJ) |
|---|---|
| ANN (L=4) | 3724.865 |
| Default IF (T=4) | 445.47 |
| PASC IF (T=4) | 446.02 |

timesteps and the end-to-end latency of sparse SNNs. Table 16 reports the inference latency of VGG-16 SNN, over the CIFAR-10 test set, for both the default IF and PASC IF configurations.

Table 16: Inference latency comparison between different IF types.

| Type of IF | Total Inference Latency (s) |
|---|---|
| Default IF (T=4) | 0.40 |
| PASC IF (T=4) | 0.45 |

## A.11 Comparison between PASCAL and direct SNN training

We compare our method against the direct training approach proposed in Deng et al. (2022). The results are presented in Table 17. We show significant improvements in accuracy across datasets, primarily for more complex datasets like ImageNet.

Table 17: Comparison between PASCAL and direct SNN training (TET).

| Dataset | Method | Model | Timesteps | Accuracy |
|---|---|---|---|---|
| CIFAR-10 | TET | ResNet-19 | 4 | 94.44 |
| | PASCAL (ours) | VGG-16 | 4.21 | 95.61 |
| CIFAR-100 | TET | ResNet-19 | 4 | 74.47 |
| | PASCAL (ours) | VGG-16 | 4.21 | 76.06 |
| ImageNet | TET | Spiking ResNet-34 | 6 | 64.79 |
| | PASCAL (ours) | ResNet-34 | 8.17 | 74.31 |

