# OpenReview forum: "PASCAL: Precise and Efficient ANN- SNN Conversion using Spike Accumulation and Adaptive Layerwise Activation"
_TMLR — Accepted by TMLR_

### Review · Reviewer_SB3M · 2025-07-25

**Summary Of Contributions:**

This paper presents a method for converting ANNs to SNNs for image classification tasks. The authors introduced a spike-count-based approach to derive an activation function that is mathematically equivalent to an ANN with QCFS activation. Moreover, the authors proposed an alternative quantization method to adaptively determine the number of quantization steps for each layer of the network, instead of using a fixed number of steps. Experimental results on standard image classification benchmarks (CIFAR-10/100 and ImageNet) and multiple architectures (VGG-16, ResNet-18/34) demonstrate that PASCAL achieves accuracy comparable to that of the original ANN. The paper also includes an estimated energy efficiency analysis, demonstrating the potential reduction in energy use as compared to ANNs.

**Audience:**

Yes

**Broader Impact Concerns:**

I do not believe this work requires a broader impact statement.

**Claims And Evidence:**

Yes

**Requested Changes:**

Please see the _Strengths and Weaknesses_ section above. Overall, I don’t have any major concerns with the paper.

**Strengths And Weaknesses:**

Strengths
- Comprehensive evaluation and comparison against existing methods for ANN-SNN conversion on computer vision models and datasets.
- The use of “inhibiting spike” to adjust the internal spike count in order to match the ANN with QCFS activation is interesting.
- The provided proofs in the Appendix seem to be sound and valid, though I did not verify them in detail.

Weaknesses
- The energy efficiency of the method is estimated, instead of measured with traditional vs neuromorphic hardware when running the ANNs and SNNs. Though this limitation is understandable.
- While the authors have compared their method against other ANN-SNN conversion methods, specifically in Table 6, I think the work can be further strengthened by adding baseline SNNs that were directly trained on the 3 vision datasets, or their spiking/neuromorphic equivalent (CIFAR10-DVS, CIRFAR100-DVS and ES-ImageNet), to demonstrate the purpose of ANN-SNN conversion.

---

> ### Author Response · Authors · 2025-08-05
> **Response to Review**
>
> 1. We agree with the reviewer that off-the-shelf hardware to support SNNs is limited. However, to address this limitation, we developed an internal SNN accelerator, which is a reimplementation of the LoAS accelerator (https://arxiv.org/html/2407.14073v1). We built an architectural simulator for the LoAS accelerator and report preliminary simulation results of energy consumed by SNN using the PASC IF layer, compared to the standard IF layer, in Appendix A.10 in the manuscript.
>
> 2. We thank the reviewer for the suggestion. We have added a table in Appendix A.11, which shows that SNNs obtained using PASCAL achieve comparable or higher accuracy compared to those obtained through direct training. We have benchmarked our work against the paper titled "Temporal Efficient Training of SNN via Gradient Re-weighting" (https://arxiv.org/abs/2202.11946).

---

### Review · Reviewer_NtDp · 2025-08-07

**Summary Of Contributions:**

This paper proposes PASCAL, a new ANN-SNN conversion framework aiming to achieve small conversion error with minimal inference timesteps. It builds on the QCFS method while proposing a spiking neuron model incorporating several stages of excitatory and inhibitory spikes, and an adaptive layerwise activation method to dynamically configure per-layer quantization steps. Experiments on several static datasets show the high performance and low latency of the method, with high estimated energy efficiency.

**Audience:**

Yes

**Broader Impact Concerns:**

No.

**Claims And Evidence:**

Yes

**Requested Changes:**

1. Properly discuss/compare the results with quantized ANNs.

2. Justify the plausibility and costs of the proposed neuronal operations on potential neuromorphic hardware.

**Strengths And Weaknesses:**

Strengths:

1. This paper shows the equivalence between converted and original models at the network level.

2. Experiments are carried out on various network architectures and datasets, showing high performance with fewer time steps compared with previous ANN-SNN conversion methods.

Weakness:

1. A longstanding problem for the ANN-SNN conversion methods is their inadequate comparison to network quantization. The proposed method is to minimize the difference from an activation quantized neural network, but no significant practical advantage is demonstrated over quantized ANNs. In the energy estimation, this paper simply assumes FP32 MAC operations for ANNs, but ANNs can actually be quantized to 8-bit or even 4-bit precision (for both weight and activation) with minimal accuracy loss, given advances in the network quantization techniques. These operations are much cheaper. Some previous work revisited SNNs from the network quantization perspective [1], offering a potential way to compare under more fair settings.

2. To minimize the difference from the activation quantized neural network, this paper introduces a lot of requirements to neuron models, such as multiple stages, both excitatory and inhibitory spikes, etc. However, there is no discussion of the plausibility and costs of them considering neuromorphic hardware that SNNs target. Actually, the development of neuromorphic hardware, such as Loihi-2, enables the support for several bits in a spike signal, akin to quantized activations. Then, why are these highly demanding techniques required, compared with directly leveraging multiple bits? The requirement of heterogeneous stages actually eliminates the potential advantage of asynchronization and parallelization of neuromorphic computing.

[1] Are Conventional SNNs Really Efficient? A Perspective from Network Quantization. CVPR 2024.

---

> ### Author Response · Authors · 2025-08-24
> **Response to Review**
>
> 1. We thank the reviewer for the suggestion to compare the energy efficiency of SNNs with quantized ANNs. Accordingly, we have updated Table 3 in the manuscript to include an energy comparison between INT8-quantized ANNs and SNNs. Our analysis shows that SNNs achieve up to 11$\times$ higher energy efficiency compared to INT8-quantized ANNs. This improvement arises because the energy consumption of an INT8 accumulate (AC) operation is roughly 7.6$\times$ lower than that of an INT8 multiply–accumulate (MAC) operation. For reference, the energy ratio of FP32 MAC to AC operations is approximately 5$\times$ in 45 nm CMOS technology [R1].
>
> In an effort to further validate the energy efficiency results, we re-implemented an ANN accelerator SparTen [R2] and SNN accelerator LoAS [R3]. Both the accelerators were designed to support INT8 quantization for the weights. In addition, SparTen was enhanced to use the QCFS function to implement quantized activation. We synthesized the Register Transfer Level (Verilog RTL) design of the accelerators using the Synopsys Design Compiler on 40 nm CMOS technology to determine the operating frequency (560 MHz) and power consumption. We then developed a cycle-accurate simulator of the respective accelerators to estimate inference latency and energy. The following Table summarizes the inference energy of VGG-16 ANN and SNN, obtained empirically from our cycle-accurate simulators, over the CIFAR-10 test set.
>
> | Type of Network | Total Inference Energy |
> | :-------------------: | :-----------------------: |
> |   ANN (L=4)          | 3724.865 mJ       |
> | Default IF (T=4)    | 445.47 mJ          |
> | PASC IF (T=4)       | 446.02 mJ          |
>
> Our results from hardware simulators, which model microarchitectural details with higher fidelity, show that SNNs achieve up to 8.4$\times$ lower inference energy compared to quantized ANNs. The energy efficiency results are reported in Appendix A.10 of the revised manuscript.
>
> 2. We would like to emphasize that the proposed PASC IF neuron adds only a minor overhead to inference latency and energy cost, since the end-to-end latency and energy consumption of the network are dominated by the MatMul layers, which incur more operations than the IF layers, as discussed in Section 4.2 and Appendix A.10. We quantify the overhead by implementing the PASC IF neuron on the LoAS accelerator [R3], which employs a fully temporal-parallel dataflow architecture designed to minimize both data movement across timesteps and the end-to-end latency of sparse SNNs. The following table reports the inference latency of VGG-16 SNN, over the CIFAR-10 test set, for both the default IF and PASC IF configurations.
>
> | Type of IF | Total Inference Latency (in seconds) |
> | :-------------------: | :-----------------------: |
> | Default IF (T=4) | 0.40 |
> | PASC IF (T=4) | 0.45 |
>
> The energy cost of the PASC IF neuron has already been reported in the previous response. Our results show that the overall energy consumption and inference latency increases by only $0.1$\% and $12.5 $\%, respectively, compared to the baseline SNN. These findings demonstrate that PASC IF operations have a limited impact on the overall parallelization efficiency of neuromorphic hardware. Moreover, PASC SNN achieves superior accuracy within a bounded number of timesteps across models and datasets (including ImageNet) compared to baseline SNN, as demonstrated both mathematically and empirically.
>
> We next discuss the feasibility of our approach on the Intel Loihi 2 neuromorphic processor [R4]. Loihi 2 supports graded spikes, using which we can transmit integer payloads up to a precision of 8 bits instead of binary spike information, at a minimal energy cost. Also, operations such as ADD, SUB, and MUL are supported on graded spikes. Therefore, such spikes can be used to support spike accumulation (used in stages 1 and 2 of PASC) as well as inhibitory spikes (used in stage 2 of PASC) without any synchronization overhead.
>
> We also update our manuscript to reflect the suggested changes.
>
> [R1] Horowitz, M., 2014, February. 1.1 computing's energy problem (and what we can do about it). In 2014 IEEE international solid-state circuits conference digest of technical papers (ISSCC) (pp. 10-14). IEEE.
>
> [R2] Gondimalla, A., Chesnut, N., Thottethodi, M. and Vijaykumar, T.N., 2019, October. SparTen: A sparse tensor accelerator for convolutional neural networks. In Proceedings of the 52nd Annual IEEE/ACM International Symposium on Microarchitecture (pp. 151-165).
>
> [R3] Yin, R., Kim, Y., Wu, D. and Panda, P., 2024, November. Loas: Fully temporal-parallel dataflow for dual-sparse spiking neural networks. In 2024 57th IEEE/ACM International Symposium on Microarchitecture (MICRO) (pp. 1107-1121). IEEE.
>
> [R4] https://download.intel.com/newsroom/2021/new-technologies/neuromorphic-computing-loihi-2-brief.pdf

---

### Review · Reviewer_xLuW · 2025-10-29

**Summary Of Contributions:**

This paper proposes a new approach for converting a neural network (NN) into a spiking neural network (SNN) with almost zero loss in performance and reduced energy consumption.
The main two contributions of the paper are:

- a spike-count and inhibition SNN, which has a formal prof of the quality of the proposed conversion. The NN is first converted into its quantized version and then into the SNN, with a fewer time-steps and therefore reduced inference time and energy consumption

- An adatptive layer-wise quantisation, in which some layers are required to be in high precision, while others can keep a lower precision, without affecting the final network performance.
The proposed approach is evaluated on standard computer vision datasets such as CIFAR-10 and 100 adn ImageNet, with VGG-16 and ResNet architectures.

**Audience:**

Yes

**Broader Impact Concerns:**

This work will allow to convert NN models into their SNN, without relevant loss in performance, but with a remarkable gain in energy consumption. This will enable tu run NN on very small and energy efficient devices. This, in turn would enable a broader expansion of AI algorithms on very constrained environnements. This could introduce an higher risk in the control and usage of these algorithms.

**Claims And Evidence:**

Yes

**Requested Changes:**

-(crititcal) I would like to see more information about inference time for the converted SNN and how they compare with previous approches.
-(critical) Is there a reason to not evaluate PASC and PASCAL on the same datasets and models?
- I would appreciate a better separation between related work and basics of the method.
- I would appreciate a bit more information about the hardware necessary to fully exploit SNN.

**Strengths And Weaknesses:**

\+ The paper is well written and motivated
\+ The paper is well organized and easy to read
\+ I appreciate the demonstration that the original NN and the SNN are mathematically equivalent, although I did not check all the proofs.
\+ Results are quite impressive with almost no-loss in accuracy in the conversion and an energy gain up-to 10x.

\- While the SNN are evaluated in terms of performance and energy consumption, I could not find any information about the inference speed of the proposed model, although in the intro the authors talk about an increased speed. Also, if the adaptation is layer-dependent, the number of time-steps can vary between layer. Thus, the final speed is based on the slowest layer, i.e. the layer with more time-steps?

\- Related work and preliminary knowledge are mixed-up in section 2. I would have appreciated two separated sections, one with related works and another one in the beginning of the methodology with the basic knowledge and formulation needed for the method.

\- Why in table 5, for the algorithm with Adaptive Layer, there are no results on ImageNet? I think it would make more sense to have the same models and datasets for the PASC and the PASCAL models.

---

> ### Author Response · Authors · 2025-11-15
> **Response to Review**
>
> **Addressing the query about inference time.**  We have quantified the inference latency of PASCAL relative to the default Integrate-Fire (IF) neuron used in prior conversion approaches in Appendix A.10.2. The results indicate that the proposed PASC neuron incurs only a minor latency overhead compared to an iso-timestep IF-based neuron. For instance, VGG16-SNN with PASC IF neuron incurs a latency of 0.45 $s$ over the CIFAR-10 testset, which is 12.5\% higher compared to that incurred by an SNN with default IF neuron (0.40 $s$). Moreover, we showed that for complex datasets such as ImageNet, the number of timesteps required by the PASC SNN to obtain the best accuracy is considerably smaller than that of standard IF-based approaches. Consequently, PASC SNN can offer comparable or even improved latency compared to previous works on complex vision datasets. In response to Reviewer NtDp's second question, we have provided additional hardware details for efficient implementation of PASC SNNs.
>
>
> **ImageNet results for PASCAL.** We have updated Table 5 of the revised manuscript with additional ImageNet results for VGG-16 and ResNet-34 SNNs, as suggested by the reviewer. This further substantiates the efficacy of PASCAL on the same datasets and models as PASC.
>
>
> **Query on  better separation between related work and basics of the method.** We thank the reviewer for the suggestion to improve the organization of the manuscript. We have moved the QCFS formalism subsection from related works to the methodology section (Section 3.1) in the revised manuscript. The related works section is now limited to discussing prior research on ANN-SNN conversion (Section 2.1) and temporally efficient SNN inference (Section 2.2).
>
>
> **Hardware.** Section A.10 in the appendix of our manuscript describes a state-of-the-art hardware accelerator for SNNs. We have quantified the inference latency and energy of PASC SNN and default IF-based SNN, and performed a comparative analysis with the baseline ANN. In addition, we have also detailed the feasibility of implementing our approach on the Intel Loihi neuromprphic chip in our response to Reviewer NtDp.

---

### Decision · Action_Editor_Ay8r · 2025-11-28

**Recommendation:** Accept as is

**Audience:**

Yes

**Audience Explanation:**

ANN-SNN conversion is an important research direction in the community of spiking neural networks, and I believe there are quite a few researchers interested in the findings of this work.

**Claims And Evidence:**

Yes

**Claims Explanation:**

The reviewers raised several questions regarding the details, and the authors have fully addressed all concerns raised by the reviewers.